# Effects of theophylline on ADCY5 activation— From cellular studies to improved therapeutic options for ADCY5-related dyskinesia patients

**Dirk Tänzler**[1,2], **Marc Kipping**[1,2], **Marcell Lederer**[3], **Wiebke F. Günther**[1,2], **Christian Arlt**[1,2], **Stefan Hüttelmaier**[3], **Andreas Merkenschlager**[4], **Andrea Sinz**[1,2]*

**1** Department of Pharmaceutical Chemistry & Bioanalytics, Institute of Pharmacy, Martin Luther University Halle-Wittenberg, Halle (Saale), Saxony-Anhalt, Germany, **2** Center for Structural Mass Spectrometry, Martin Luther University Halle-Wittenberg, Halle (Saale), Saxony-Anhalt, Germany, **3** Institute of Molecular Medicine, Section for Molecular Cell Biology, Faculty of Medicine, Martin Luther University Halle-Wittenberg, Halle, Saxony-Anhalt, Germany, **4** Department of Neuropediatrics, Hospital for Children and Adolescents, University of Leipzig, Leipzig, Saxony, Germany

* andrea.sinz@pharmazie.uni-halle.de

## Abstract

We show the effects of the three purine derivatives, caffeine, theophylline, and istradefylline, on cAMP production by adenylyl cyclase 5 (ADCY5)-overexpressing cell lines. A comparison of cAMP levels was performed for ADCY5 wild-type and R418W mutant cells. ADCY5-catalyzed cAMP production was reduced with all three purine derivatives, while the most pronounced effects on cAMP reduction were observed for ADCY5 R418W mutant cells. The gain-of-function ADCY5 R418W mutant is characterized by an increased catalytic activity resulting in elevated cAMP levels that cause kinetic disorders or dyskinesia in patients. Based on our findings in ADCY5 cells, a slow-release formulation of theophylline was administered to a preschool-aged patient with ADCY5-related dyskinesia. A striking improvement of symptoms was observed, outperforming the effects of caffeine that had previously been administered to the same patient. We suggest considering theophylline as an alternative therapeutic option to treat ADCY5-related dyskinesia in patients.

## Introduction

Adenylyl cyclases (ADCYs) are central enzymes in all organisms [1]. The ADCY family comprises ten isoforms; nine of them are membrane-bound enzymes [2, 3]. ADCY5, which is most commonly expressed in brain and heart tissue, is one of the least studied isoforms [3]. As with all ADCYs, ADCY5 converts adenosine triphosphate (ATP) to cyclic adenosine-3',5'-monophosphate (cAMP) and pyrophosphate (Fig 1) [3]. ADCY5 has been identified as the primary ADCY isoform that is responsible for up to 80% of the cAMP production in striatal medium spiny neurons [3, 4]. The complex system of initiating and controlling movement is dependent on a well-balanced signaling between G-Protein-Coupled-Receptors (GPCRs) and ADCYs [4]. Different neurotransmitters stimulate or inhibit hydrolysis of ATP to cAMP via ADCY-coupled GPCRs [5]. As such, adenosine receptor 2A ($A_{2A}$) agonists increase intercellular cAMP levels [5].

**Data Availability Statement:** All relevant data are within the paper and its Supporting Information files.

**Funding:** AS acknowledges financial support by the DFG (www.dfg.de) (RTG 2467, project number 391498659 "Intrinsically Disordered Proteins – Molecular Principles, Cellular Functions, and Diseases", RTG 2751 "InCuPanC", project number 449501615, INST 271/404-1 FUGG, INST 271/405-1 FUGG, and CRC 1423, project number 421152132), the Federal Ministry for Economic Affairs and Energy (www.bmwk.de) (BMWi, ZIM project KK5096401SK0), the region of Saxony-Anhalt (www.sachsen-anhalt.de), and the Martin Luther University Halle-Wittenberg (www.uni-halle.de) (Center for Structural Mass Spectrometry). The funders had no role in study design, data collection and analysis, decision to publish, or preparation of the manuscript.

**Competing interests:** The authors have declared that no competing interests exist.

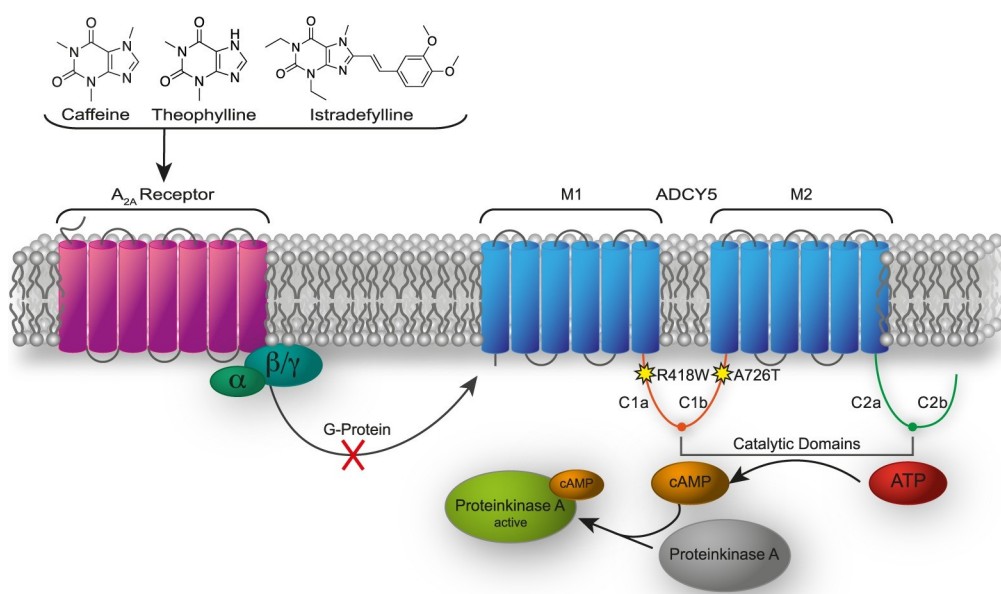

**Fig 1. Mode of action for caffeine, theophylline, and istradefylline on the reduction of cAMP production by ADCY5 via inhibiting the A2A receptor.**

ADCY5 contains an intracellular N-terminal domain, two membrane domains (M1 and M2) consisting of six helices, two cytoplasmic homologous domains (C1 and C2), and a short intracellular loop between the two catalytic domains (Fig 1) [2, 3, 6]. The formation of an ATP-binding pocket that brings the C1 and C2 domains into proximity is crucial for ADCY5's catalytic activity is [2, 3]. Amino acid mutations in ADCY5 at different positions within the protein have been described, with the most prominent ones being R418W, R418G, R418Q and A726T [2, 4, 7]. The A726T mutant, in which an alanine is replaced by a threonine residue (Fig 1) in the C1b domain, influences ADCY5's flexibility. This amino acid exchange, resulting from a single-point mutation, is suggested to alter the structure of the ATP-binding pocket by affecting the interaction strength between the C1 and C2 domains [2, 3]. The most frequently occurring variant of ADCY5 has a R418W exchange (Fig 1), where a positively-charged amino acid residue with a relatively long aliphatic side-chain (arginine) is replaced by a hydrophobic, aromatic amino acid residue (tryptophan). This exchange also results from a single-point mutation and is considered to alter the response as well as transmission of β-adrenergic receptor stimulation [2].

Gain-of-function ADCY5 mutants cause a significant increase in the catalytic hydrolysis of ATP in response to adrenergic stimulation [4, 8], resulting in increased intracellular cAMP concentrations in striatal cells [8]. All ADCY5 mutations result in kinetic disorders or dyskinesia that effect all muscles [2, 4, 9]. A combination of linkage analysis and exon sequencing provided strong evidence for mutations of ADCY5 being responsible for Familial Dyskinesia and Facial Myokymia (FDFM) [2, 3].

The inhibition of ADCYs impacts the conversion of ATP to cAMP in the gain-of-function mutants and therefore reduces intracellular cAMP concentrations [10, 11]. Increasing evidence suggests that A2A receptor antagonists exert positive effects on the symptoms of ADCY5-related dyskinesia [8]. Caffeine is an A2A antagonist that is hypothesized to alter cAMP levels in striatal neurons [12]. Theophylline showed a higher potency regarding different metabolic effects compared to caffeine [13], as well as resulting in improved ergogenic effects during whole body exercise [13]. Very recently, a report has been published that

provides insight into the treatment of ADCY5-related dyskinesia patients with caffeine and which describes the effects on frequency and duration of paroxysmal movement disorders, baseline movement disorders, and other motor and no-motor features [14]. Overall, a consistent quality-of-life improvement was reported in 87% of patients. Only three out of 30 patients of this retrospective study reported a worsening of symptoms [14]. Based on these findings, the use of caffeine has been suggested as first-line treatment of ADCY5-related dyskinesia [14].

In the work reported here, we compare the effects of the three purine derivatives, caffeine [14], theophylline, and istradefylline [8], regarding their reduction of cAMP levels in ADCY5-overexpressing cell lines (ADCY5 wild-type and R418W mutant). We give a rationale for administering theophylline to ADCY5-related dyskinesia patients based on the highly promising results obtained for one patient. We envision that theophylline has the potential to complement or even replace the therapy with caffeine for treating patients with ADCY5-related dyskinesia.

## Materials and methods

### Chemicals and reagents

All chemicals and reagents were obtained from Roth or Sigma Aldrich at the highest purity available.

### Cell culture

HEK293T/17 cells (ATCC, RRID: CVCL_1926) were stably transfected with vectors encoding GFP (for comparative purposes only), ADCY5 wild-type (ADCY5wt), and ADCY5 R418W mutant (ADCY5mut). Cells were cultured in Dulbecco's modified Eagle's Medium (DMEM) supplemented with 10% (v/v) fetal bovine serum (FBS) for two days in six-well plates at 37°C and 5% $CO_2$; 4.4 x $10^5$ cells were cultivated in one well. Each treatment was performed in triplicate. Cells were incubated with caffeine, theophylline, and istradefylline at 37°C and 5% $CO_2$ using different concentrations (caffeine and theophylline: 1 μM, 10 μM, 100 μM, 1 mM; istradefylline: 1 nM, 10 nM, 100 nM, 1 μM) for different time periods (10, 30, 60, 120, 240, and 480 min). Time-course experiments were performed in six replicates. Cells were harvested with PBS buffer, washed, and centrifuged at 1,500 x $g$ for 5 min. A 200 μl aliquot of trizol reagent (9.5 g guanidinium thiocyanate, 3.1 g ammonium thiocyanate, 3.5 ml of 3 M sodium acetate, 5 g glycerol, 48 ml Roti Aqua-Phenol in 100 ml total volume) was added to the cell pellets to facilitate cell lysis and at the same time inactivate cAMP-degrading enzymes. Non-lysed cells and cell debris were removed by a centrifugation step using 30-kDa molecular weight cut-off filters (Amicon Millipore), which was performed at 14,000.0 x $g$ for 10 minutes. Filtrates were stored at 4°C until they were analyzed by LC-MS/MS.

### Liquid Chromatography-Tandem Mass Spectrometry (LC-MS/MS)

LC separation of nucleotides (cAMP, AMP, ATP, cGMP, GMP, GDP, and GTP) was performed with a UPLC I-Class FTN system (Waters) equipped with an Atlantis Premier BEH C18 AX column (2.1 mm x 50 mm, 1.7μm, Waters). Separation was performed at a flow rate of 400 μl/min with the following LC gradient: 0–1 min: 3% B, 1–4 min: 3–20% B, 4–5 min: 20–85% B, and 5–5.5 min: 85% B; solvent A, 0.2% (v/v) formic acid in water; solvent B, 0.2% (v/v) formic acid in acetonitrile. The UPLC system was directly coupled to a Xevo TQ-XS mass spectrometer (Waters) with an electrospray ionization (ESI) source. Multiple-reaction monitoring (MRM) was performed using specific transitions for the nucleotides cAMP, AMP, ATP, cGMP, GMP, GDP, and GTP.

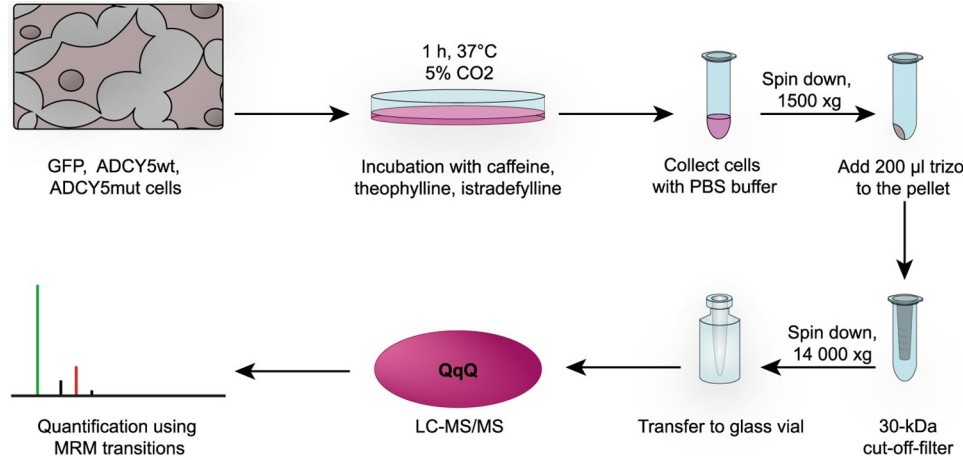

**Fig 2. Analytical workflow.**

### Ethics statement

The ethics committee of the University of Leipzig was informed and approved this study. The study has been registered at the DRKS (ID: DRKS00029154). Written consent was obtained from the parents.

## Results

GFP, ADCY5wt, and ADCY5mut cells were compared for their ability to catalyze ADCY5-dependent cAMP production according to the workflow presented in Fig 2. Cells were incubated with caffeine, theophylline, and istradefylline at varying concentrations and samples were collected at different time points. All three drugs are purine derivatives that exert their effects by binding to the $A_{2A}$ receptor, resulting in inhibition of ADCY5 and reduced cAMP levels (Fig 1). As ADCY5-related dyskinesia is caused by gain-of-function ADCY5 amino acid-exchange mutations (Fig 1), reduction of cAMP levels will result in an improvement in movement disorders. This effect of caffeine on ADCY5-related dyskinesia patients has been impressively demonstrated in a recent report [14].

The main goal of the present study was to compare the effects of the three purines, caffeine, theophylline, and istradefylline, on cAMP concentrations at the cellular level. Caffeine, theophylline, and istradefylline differ in their $K_i$ values (caffeine 23.4 μM, theophylline 1.7 μM, istradefylline 9.2 nM) at the $A_{2A}$ receptor and should therefore exhibit different effects on the mediated ADCY5-catalyzed cAMP production (Fig 1). After harvesting and lysing the cells, filtrates were collected and analyzed by LC-MS/MS (Fig 2). Based on MRM analyses, cAMP concentrations were quantified after treatment with the three purine drugs using cells that stably express GFP (control), ADCY5 wild-type, and ADCY5 R418W mutant. This allowed us to gain detailed insights into the ability of caffeine, theophylline, and istradefylline to reduce cAMP at the cellular level.

### Comparison of caffeine, theophylline, and istradefylline regarding the reduction of cAMP levels

Consistent with the pivotal role of ADCY5 for cAMP production and its substantially increased catalytic activity caused by the R418W mutation, basal cAMP levels were upregulated ~5-fold in ADCY5wt while they were up-regulated ~30-fold in ADCY5mut cells. Effects

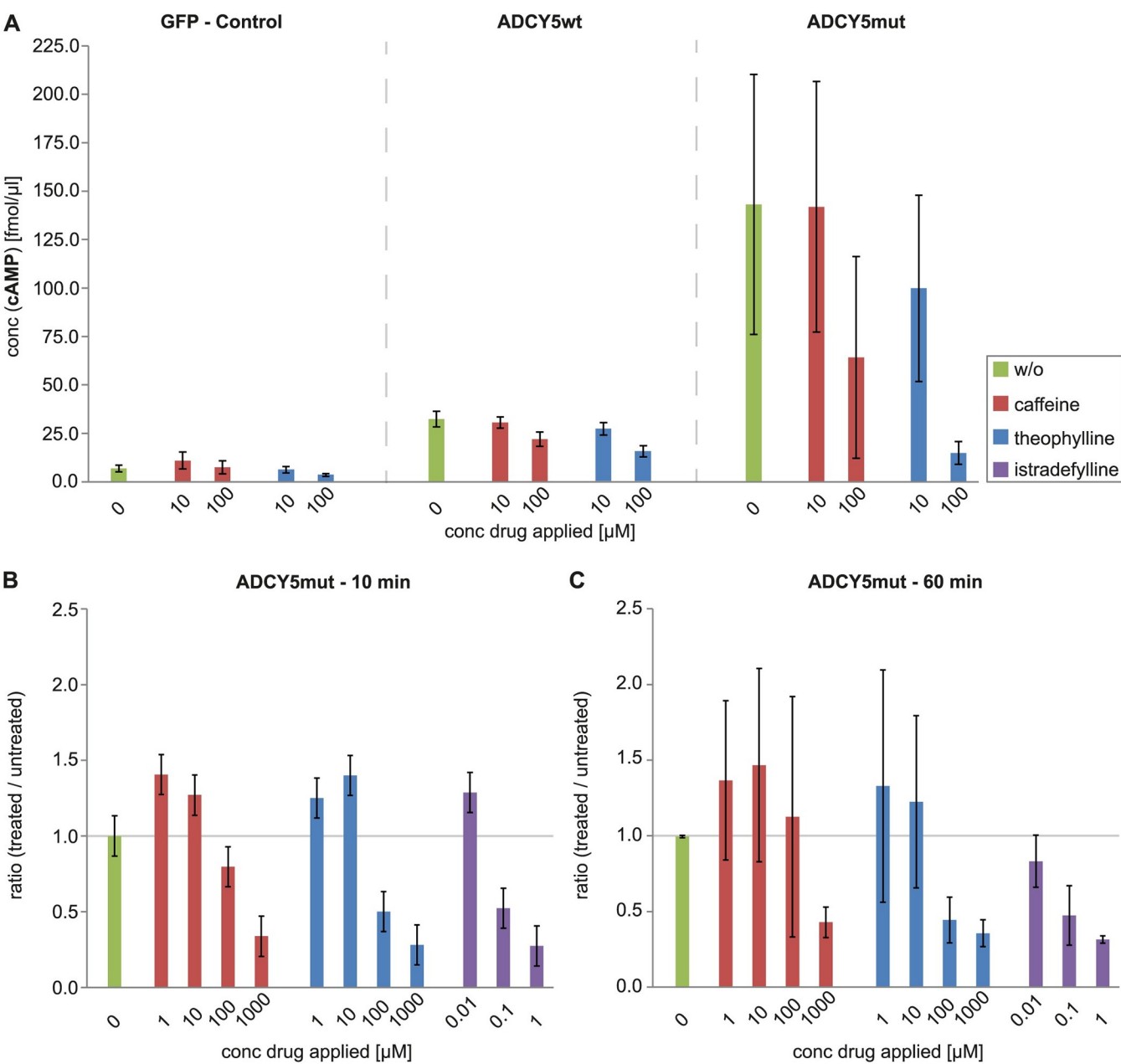

**Fig 3. Influence of caffeine (red bars), theophylline (blue bars), and istradefylline (purple bars) at different concentrations on cAMP levels of different cell lines at different time points.** Controls without drug treatment are shown as green bars. (**A**) GFP, ADCY5wt, and ADCY5mut cells, 1 h treatment. cAMP concentration of GFP cells (without ADCY5) are set to a value of 1.0. All cAMP concentrations of ADCY5mut cells are related to the GFP reference. (**B**) ADCY5mut cells, 10 min drug treatment (**C**) ADCY5mut cells, 1 h drug treatment. Experiments were performed in 3 replicates.

of the three purines were found to be most pronounced for ADCY5mut cells, showing a drastic reduction of cAMP levels, even at concentrations of 100 µM caffeine and theophylline (Fig 3A). Already after ten minutes of drug treatment, initial reductions in cAMP levels were observed (Fig 3B); after one hour of drug treatment the effects on lowering cAMP concentrations became even more pronounced (Fig 3C). Interestingly, the cAMP concentrations observed after treatment of ADCY5mut cells with 100 µM theophylline attained roughly the same level as for non-treated ADCY5wt and GFP cells (Fig 3A). Apparently, treatment of

ADCY5mut cells with a high concentration of theophylline results in basal cAMP levels that cannot be reduced any further. At the same time, rather large variations in cAMP levels were observed between different replicates in ADCY5mut cells (Fig 3A). This might be explained by the fact that the gain-of-function ADCY5 R418W mutant exhibits up to 30-fold increased basal cAMP levels compared to ADCY5 wild-type protein.

A comparison between the three purines under investigation revealed the most prominent reduction of cAMP concentrations in ADCY5mut cells for istradefylline, followed by theophylline and caffeine after already 10 minutes (Fig 3B). Apparently, the reduction of cAMP concentration correlates well with the respective $K_i$ values (see above) of the three purines at the $A_{2A}$ receptor. After one-hour of treatment of the ADCY5mut cells with caffeine, theophylline, and istradefylline (Fig 3C), the reduction of cAMP concentrations essentially exhibited the same profile as that observed at 10 minutes of drug treatment.

## Determination of cAMP concentration by LC-MS/MS

To determine ADCY5 activity, cAMP concentrations were determined by LC-MS/MS, quantifying different nucleotides by an LC-MRM-MS/MS approach to rule out any misassignments. Specific transitions were considered for cAMP, AMP, ATP, cGMP, GMP, GDP, and GTP (Fig 4A). For cAMP identification and quantification, the MS signal response showed a linear behavior in a concentration range up to 10 pmol/µl (Fig 4B). AMP, cAMP, GMP, cGMP reference compounds were used at a concentration of 100 fmol each, showing an LC baseline separation of all nucleotides with a total elution time of 3.5 minutes (Fig 4C). Based on our

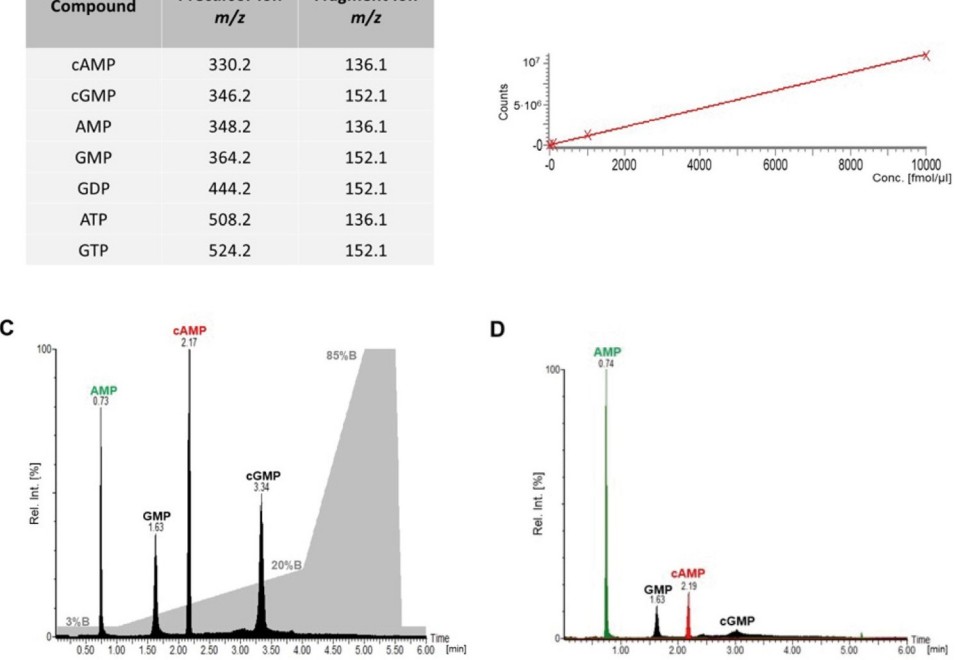

**Fig 4.** (**A**) Selected transitions in triple-quadrupole MS/MS measurements for selected nucleotides. (**B**) Linearity of mass spectrometric response for cAMP in the concentration range between 0–10 pmol/µl. (C) LC-MRM-MS/MS analysis (100 fmol) of four selected nucleotides (AMP, cAMP, GMP, cGMP). The TIC (Total Ion Current) of the LC elution is shown in black, the LC gradient is shown in grey (%B). (**D**) LC-MRM-MS/MS of filtrates (see Fig 2) for selected nucleotides.

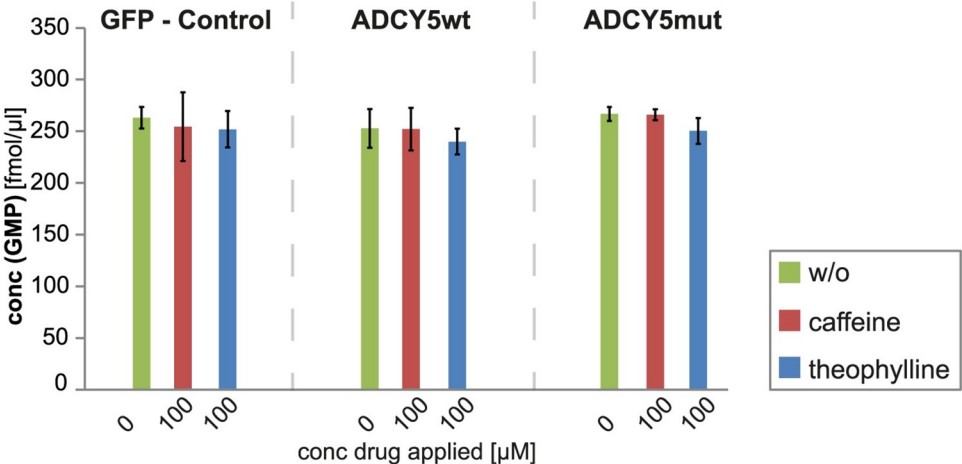

**Fig 5. Influence of caffeine (red bars) and theophylline (blue bars) at a concentration of 100 µM on GMP levels of different cell lines, 1 h drug treatment.** Controls without drug treatment are shown as green bars. Experiments were performed in 3 replicates.

LC-MRM-MS/MS method, reliable and unambiguous cAMP quantification was performed using the cells' supernatants (Fig 4D). As trizol reagent was used to stop all cellular reactions in our workflow, PDEs that are eventually present will also be inhibited during our analytical workflow (Fig 2), resulting in a lack of cAMP-to-AMP conversion.

To confirm that an inhibition of PDEs by purines does not contribute to altered cAMP levels, we also measured GMP levels after treatment of cells with 100 µM caffeine and theophylline (Fig 5). Clearly, GMP levels do not change, verifying that PDEs are not involved in converting cGMP to GMP. It has to be noted in this context that our cGMP detection limit is at ~1 fmole. As we did not detect cGMP in any of our LC-MS/MS analyses, this again confirms that PDEs are not inhibited by the purines as this would result in an accumulation of cGMP. We are therefore confident that the cAMP concentrations determined by our workflow (Fig 2) present a correct reflection of how the three purines regulate ADCY5 activity via $A_{2A}$ receptor inhibition in the cellular systems studied herein.

### Time course of caffeine and theophylline treatment on ADCY5mut cells

Treatment of ADCY5mut cells with caffeine and theophylline (10 µM and 100 µM, each) over a time-course of 480 minutes revealed a reduction in cAMP levels at both concentrations (Fig 6). Strikingly, the reduction in cAMP levels was slightly more pronounced for theophylline compared to caffeine.

### Theophylline treatment of a patient with ADCY5-related dyskinesia

Given the promising results of reducing cAMP concentrations by theophylline, especially in ADCY5mut cells, we hypothesized that theophylline might be an alternative to caffeine for treatment of patients with ADCY5-related dyskinesia [14]. Theophylline is available as a slow-release formulation and has been shown to exhibit only minor side effects in children with status asthmaticus that were treated with a low-dose of theophylline (5–7 mg/kg per day) [15]. Administering theophylline as a slow-release formulation might be favorable compared to caffeine and result in a more efficient reduction of cAMP levels in patients [16, 17]. Theophylline

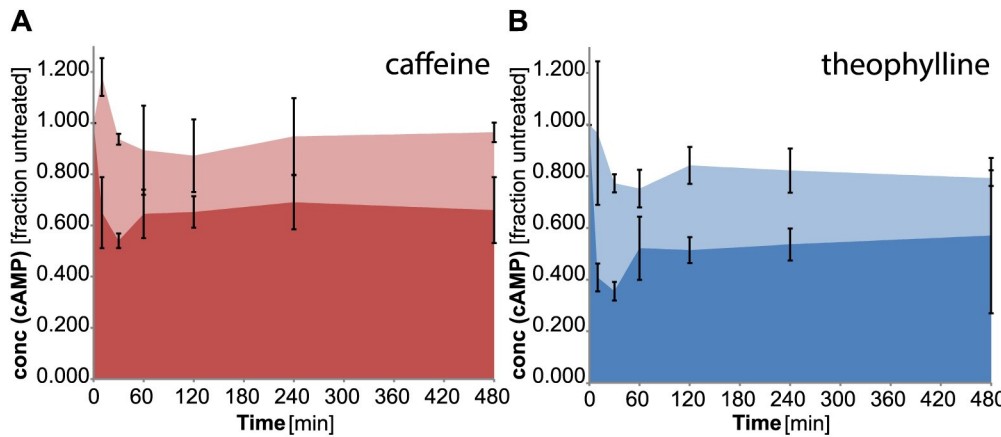

**Fig 6.** Time-dependence of (**A**) caffeine (10 μM: light red, 100 μM: red) and (**B**) theophylline (10μM: light blue, 100μM: blue) treatment of ADCY5mut cell lines. cAMP concentrations were monitored over a time-range of 8 hours. cAMP concentration of GFP cells (without ADCY5) was set to 1.0. All cAMP concentrations measured for ADCY5mut cells are related to the GFP reference. Experiments were performed in 6 replicates.

exhibits a narrow therapeutic window, which however can be efficiently met by a careful up-dosing of the drug and by regularly monitoring relevant blood parameters (see S1 Appendix).

A 6-year old female patient with ADCY5-related dyskinesia that had been successfully treated with 150 mg of caffeine (3x 50 mg daily) was treated with 400 mg of slow-release theophylline (2x 200 mg daily). The patient was diagnosed in 2018 with ADCY5 variant 1252 C>T, resulting in Arg418Trp, heterozygous and de novo mutation, no mosaicism. Initially, theophylline and caffeine were given in combination, but during the course of the therapy, the daily caffeine dose was reduced over a period of five months until theophylline was administered alone. The theophylline dose was gradually increased from ~6 mg/kg/day to ~23 mg/kg/day. After 6–8 weeks, theophylline blood levels ranged between 7.5 mg/l to 21.1 mg/l.

Already at a theophylline dose of ~12 mg/kg/day, divided into two single doses, the following effects were observed: The patient became more upright, showed increased muscle tone, was able to stand independently, and walk up to 50 meters without help. With caffeine alone, neither independent standing nor walking was possible. According to the physicians' report during therapy, the patient was able to make less than 10 steps (with assistance only) and was in a wheelchair during the day. Targeted, slow gripping was possible. Speech was impaired as the patient could speak four to five words and was hard to understand.

Under theophylline treatment, the physicians' report states an improvement in the quality of sleep as dyskinetic movements subsided completely when falling asleep. At the same theophylline dose (~12 mg/kg/day), the paroxysmal dyskinesia continued to fluctuate and recurrent administration to the patient led to a significant reduction of every symptom, such as the loss of independent standing. Bacterial or viral infection-associated deterioration of symptoms (see below) persisted despite increasing the daily dose of slow-release theophylline to ~18 mg/kg/day.

By increasing the theophylline dose to ~23 mg/kg/day, continuous improvements were observed. Supported by foot orthoses, the patient was able to walk ~50 meters hand-held and 7 meters freely, and climbed stairs independently. The paroxysmal dyskinesia showed a decrease in frequency and duration, and significantly improved dysarthria and functional hypersalivation.

On a scale from 0 (no improvement) to 10 (no symptoms), theophylline had a more pronounced influence in the afternoon (6–7), while its influence was ranked between 3–4 in the

morning. The rating was done by the medical staff after consultation with the patient's parents. The duration and frequency of episodes cannot be reliably quantified as different causes (light bacterial or viral infections, stress, joy, etc.) influence motoric conditions during theophylline treatment.

We did not observe any adverse side effects during theophylline dose titration; arterial blood pressure, pulse rate, and theophylline-related laboratory parameters were within the given reference ranges. Even at the maximum dose of ~23 mg/kg/day, sleep quality was greatly improved with no interruption of sleep for ten hours. Several patients are now being treated worldwide with theophylline. In none of the cases, side effects are observed for theophylline when the dosage is slowly increased as described in our dosage scheme and blood parameters are monitored regularly as described (see S1 Appendix).

## Conclusions

We show that cAMP levels in ADCY5-overexpressing cells were reduced with all three purines, caffeine, theophylline, and istradefylline, under investigation. The effects were most prominent for the gain-of-function ADCY5mut cell line (R418W) and were in agreement with the Ki values of the three purines at the $A_{2A}$ receptor: The most prominent reduction of cAMP levels was observed for istradefylline, followed by theophylline and caffeine showing a less pronounced cAMP reduction. As istradefylline exhibits side-effects even at low-dose application it is currently not an approved drug in Germany [18]. Theophylline, on the other hand, is available as a slow-release formulation, showing only minor side-effects, even in pediatric applications despite its narrow therapeutic window. Theophylline was therefore administered as a slow-release formulation to a preschool-aged patient with ADCY5-related dyskinesia. Independent standing and walking, as well as sleep quality were substantially improved when treating the patient with theophylline compared to caffeine. We are aware that the cellular system used herein does not reflect the situation in the striatum. However, given the outstanding results in the patient described in this study—as well as in other patients that are being treated with theophylline worldwide now—this impressive improvement of dyskinesia underlines the results of our cellular studies. Based on our findings, the efficacy and safety profile of theophylline should now be evaluated for a larger patient cohort.

## Supporting information

**S1 Appendix. Dosage scheme for theophylline as slow-release formulation.**
(DOCX)

**S1 Dataset. LC/MS raw data underlying Figs 3 and 5.** The folder names (Fig 3A, Fig 3BC, and Fig 5) reflect the respective figure numbers in the main manuscript. The zip-file includes raw data, TargetLynx data and excel files of quantitation analysis.
(ZIP)

## Acknowledgments

Dr. Wendy H. Raskind, University of Washington, Seattle, WA, is acknowledged for providing the ADCY5 plasmid. Dr. Frank Erdmann, Martin Luther University Halle-Wittenberg, is acknowledged for initial discussions; the Waters Corp. is acknowledged for providing the Xevo TQ-XS mass spectrometer. The authors are indebted to Prof. Gary Sawers, Martin Luther University Halle-Wittenberg, for critical reading of the manuscript.

## Author Contributions

**Conceptualization:** Andrea Sinz.

**Data curation:** Marc Kipping.

**Formal analysis:** Dirk Tänzler, Marc Kipping, Andrea Sinz.

**Funding acquisition:** Andrea Sinz.

**Investigation:** Andreas Merkenschlager.

**Methodology:** Dirk Tänzler, Marc Kipping, Marcell Lederer, Andrea Sinz.

**Project administration:** Andrea Sinz.

**Resources:** Marcell Lederer, Stefan Hüttelmaier, Andrea Sinz.

**Supervision:** Stefan Hüttelmaier, Andreas Merkenschlager, Andrea Sinz.

**Validation:** Andrea Sinz.

**Visualization:** Christian Arlt.

**Writing – original draft:** Wiebke F. Günther, Andrea Sinz.

**Writing – review & editing:** Andrea Sinz.

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
