## [Decision Letter · Decision Letter 0]

14 Nov 2022

PONE-D-22-23902

Effects of Theophylline on ADCY5 Activation - From Cellular Studies to Improved Therapeutic Options for ADCY5-Related Dyskinesia Patients

PLOS ONE

Dear Dr. Sinz,

Thank you for submitting your manuscript to PLOS ONE. After careful consideration, we feel that it has merit but does not fully meet PLOS ONE’s publication criteria as it currently stands. Therefore, we invite you to submit a revised version of the manuscript that addresses the points raised during the review process.

Please note that we have only been able to secure a single reviewer to assess your manuscript. We are issuing a decision on your manuscript at this point to prevent further delays in the evaluation of your manuscript. Please be aware that the editor who handles your revised manuscript might find it necessary to invite additional reviewers to assess this work once the revised manuscript is submitted. However, we will aim to proceed on the basis of this single review if possible.

The reviewer has raised major concerns about the conclusions not being fully supported by the data. The manuscript should therefore be thoroughly revised, with interpretations fully supported by the data, and the limitations discussed in full. 

We look forward to receiving your revised manuscript.

Kind regards,

Alice Coles-Aldridge

Editorial Office

PLOS ONE

Journal Requirements:

2. Please ensure that you have specified (1) whether consent was informed and (2) what type you obtained (for instance, written or verbal, and if verbal, how it was documented and witnessed). If your study included minors, state whether you obtained consent from parents or guardians. If the need for consent was waived by the ethics committee, please include this information.

“AS acknowledges financial support by the DFG (RTG 2467, project number 391498659 “Intrinsically Disordered Proteins – Molecular Principles, Cellular Functions, and Diseases”, RTG 2751 “InCuPanC”, project number 449501615, INST 271/404-1 FUGG, INST 271/405- 1 FUGG, and CRC 1423, project number 421152132), the Federal Ministry for Economic Affairs and Energy (BMWi, ZIM project KK5096401SK0), the region of Saxony-Anhalt, and the Martin Luther University Halle-Wittenberg (Center for Structural Mass Spectrometry).”

“AS acknowledges financial support by the DFG (www.dfg.de) (RTG 2467, project number 391498659 “Intrinsically Disordered Proteins – Molecular Principles, Cellular Functions, and Diseases”, RTG 2751 “InCuPanC”, project number 449501615, INST 271/404-1 FUGG, INST 271/405-1 FUGG, and CRC 1423, project number 421152132), the Federal Ministry for Economic Affairs and Energy (www.bmwk.de) (BMWi, ZIM project KK5096401SK0), the region of Saxony-Anhalt (www.sachsen-anhalt.de), and the Martin Luther University Halle-Wittenberg (www.uni-halle.de) (Center for Structural Mass Spectrometry).

Reviewers' comments:

Reviewer's Responses to Questions

**Comments to the Author**

1. Is the manuscript technically sound, and do the data support the conclusions?

Reviewer #1: No

2. Has the statistical analysis been performed appropriately and rigorously? 

Reviewer #1: N/A

3. Have the authors made all data underlying the findings in their manuscript fully available?

Reviewer #1: Yes

4. Is the manuscript presented in an intelligible fashion and written in standard English?

Reviewer #1: Yes

5. Review Comments to the Author

Reviewer #1: In this paper, Tänzler et al. report the results of theophylline and caffeine administration on cAMP concentrations in WT and ADCY5 mutant cells, as well as the results of theophylline treatment in a ADCY5 patient. It is overall well-written and easy to read. Even though both the cellular data and clinical report are of interest, there are issues with the way data are interpreted and what the authors conclude.

1) The first issue concerns the in vitro model that is used: both a non-neuronal (HEK) and non-integrated (cells) model. Given the complexity of striatal function, this limits greatly what can be inferred from the results – which should be mentioned in text.

2) The authors draw conclusions concerning caffeine and theophylline efficacy on cAMP lowering (“most prominent reduction of cAMP concentration for istradefylline, followed by theophylline and caffeine”, “strikingly, the reduction in cAMP level was more pronounced for theophylline compared to caffeine”) that are not supported by the data they present. Indeed, the same concentrations have been used for both molecules even though they are not equivalent, and the differences are clearly not statistically significant. This need to be rephrased, and limitations need to be clearly presented.

3) The authors present theophylline as a safe treatment, choosing to cite a paper reporting only minor side effects but at very low doses (5-7 mg/kg per day), even though they treat their ADCY5 patient with 23 mg/kg per day, and they do not cite any report on theophylline toxicity. Yet therapeutic window is known to be extremely narrow with this molecule, and there are several reports of sometimes fatal theophylline toxicity on the literature. So much so that theophylline is not prescribed for asthma if other medications are available (https://www.ncbi.nlm.nih.gov/books/NBK532962/). Such toxicity is only reported with extremely high doses of caffeine, making it the safest treatment of the two, which should be said in text. What’s more, the authors sate that theophylline was more effective than caffeine, but they used comparatively much higher doses of theophylline than caffeine in this patient. Even though doses up to 800 mg/day have been used in the recent report cited by the authors, they seem to have stopped at 150 mg a day in their patient. Have higher doses been tried? Have side effects prevented to raise the dose? In all likelihood, the benefits observed from high-dose theophylline treatment would have been observed with higher doses of caffeine, so that concluding here to the superiority of theophylline is quite misleading (the only advantage that could be put forward here is its slow-release formulation).

The case report part should therefore be rewritten to make room for all this additional (and crucial) information.

4) The authors state in the conclusion that istradefylline exhibits severe side effects. What are their sources? Most reports seem to conclude to limited side effects and acceptable tolerance with this drug. Most reported side effects appear to be dyskinesia, confusion and hallucinations, which are common in a population of Parkinson’s disease patients but cannot be generalized to other patients. Therefore, this needs to be rephrased.

6. PLOS authors have the option to publish the peer review history of their article (what does this mean?). If published, this will include your full peer review and any attached files.

Reviewer #1: No

---

## [Author Response · Author response to Decision Letter 0]

25 Nov 2022

Responses to Reviewer’s Comments

Question: In this paper, Tänzler et al. report the results of theophylline and caffeine administration on cAMP concentrations in WT and ADCY5 mutant cells, as well as the results of theophylline treatment in a ADCY5 patient. It is overall well-written and easy to read. Even though both the cellular data and clinical report are of interest, there are issues with the way data are interpreted and what the authors conclude.

Response: We would like to thank this reviewer for his/her overall favorable comments on our work. I would like to point out that we performed all experiments in a diligent and careful manner and we believe that our conclusions are fully justified.

Question: The first issue concerns the in vitro model that is used: both a non-neuronal (HEK) and non-integrated (cells) model. Given the complexity of striatal function, this limits greatly what can be inferred from the results – which should be mentioned in text.

Response: We are aware that the cellular system used herein does not reflect the situation in the striatum. However, given the outstanding results in the patient described in this study (as well as in nine further patients treated with theophylline worldwide now) this impressive improvement of dyskinesia confirms the results of our cellular studies. We added a sentence addressing this issue in the “Conclusions” part on page 15: “We are aware that the cellular system used herein does not reflect the situation in the striatum. However, given the outstanding results in the patient described in this study - as well as in other patients treated with theophylline worldwide now - this impressive improvement of dyskinesia confirms the results of our cellular studies.”

Question: The authors draw conclusions concerning caffeine and theophylline efficacy on cAMP lowering (“most prominent reduction of cAMP concentration for istradefylline, followed by theophylline and caffeine”, “strikingly, the reduction in cAMP level was more pronounced for theophylline compared to caffeine”) that are not supported by the data they present. Indeed, the same concentrations have been used for both molecules even though they are not equivalent, and the differences are clearly not statistically significant. This need to be rephrased, and limitations need to be clearly presented.

Response: We agree with the reviewer that according to Figure 3B, the effects of cAMP reduction for caffeine and theophylline are similar to each other after 10 minutes of drug treatment. However, considering the effects of caffeine and theophylline at 60 minutes after drug treatment, the effect of theophylline in respect to cAMP level reduction is more pronounced at a concentration of 100 µM. We rephrased the respective sentences in the text: “The most prominent reduction of cAMP levels was observed for istradefylline, followed by theophylline and caffeine showing a less pronounced cAMP reduction.” “Strikingly, the reduction in cAMP levels was slightly more pronounced for theophylline compared to caffeine.”

Question: The authors present theophylline as a safe treatment, choosing to cite a paper reporting only minor side effects but at very low doses (5-7 mg/kg per day), even though they treat their ADCY5 patient with 23 mg/kg per day, and they do not cite any report on theophylline toxicity. Yet therapeutic window is known to be extremely narrow with this molecule, and there are several reports of sometimes fatal theophylline toxicity on the literature. So much so that theophylline is not prescribed for asthma if other medications are available (https://www.ncbi.nlm.nih.gov/books/NBK532962/). Such toxicity is only reported with extremely high doses of caffeine, making it the safest treatment of the two, which should be said in text. 

Response: We are fully aware of the narrow therapeutic window of theophylline, which however can be efficiently met by a careful up-dosing of the drug. The reason why theophylline is not being used as first-line treatment for asthma bronchiale any more is that the development of ß2-sympathomimetics has advanced tremendously in the last 20 years. These drugs, such as salbutamol, salmeterol etc, exhibit a better efficacy for treating asthma bronchiale as theophylline.

During the last months, I have been approached by a number of parents of children suffering from ADCY5-related dyskinesia and I am advising neurologists and pediatricians on how to safely use theophylline. For this, I have assembled a guideline for medical doctors (and patients) regarding the dosage of theophylline as slow-release formulation for treating patients with ADCY5-related dyskinesia. We have now included this guideline in the Supporting Information of this manuscript.

Due to the spectacular success of theophylline treatment in the patient described in our manuscript who was confined to a wheelchair and is now able to walk without help, nine children with ADCY5-related dyskinesia are now being treated worldwide with theophylline. The videos the parents share with me are partially breathtaking. In none of the cases, side effects are observed for theophylline when the dosage is slowly increased as described in our dosage scheme and blood parameters are monitored regularly as described (see Supporting Information).

To clarify this important point, two sentences were added on pages 13, 14, and 15: “Theophylline exhibits a narrow therapeutic window, which however can be efficiently met by a careful up-dosing of the drug and by regularly monitoring relevant blood parameters (see Supporting Information).” “Several patients are now being treated worldwide with theophylline. In none of the cases, side effects are observed for theophylline when the dosage is slowly increased as described in our dosage scheme and blood parameters are monitored regularly as described (see Supporting Information).” “Theophylline, on the other hand, is available as a slow-release formulation, showing only minor side-effects, even in pediatric applications despite its narrow therapeutic window.”

Question: What’s more, the authors sate that theophylline was more effective than caffeine, but they used comparatively much higher doses of theophylline than caffeine in this patient. Even though doses up to 800 mg/day have been used in the recent report cited by the authors, they seem to have stopped at 150 mg a day in their patient. Have higher doses been tried? Have side effects prevented to raise the dose? In all likelihood, the benefits observed from high-dose theophylline treatment would have been observed with higher doses of caffeine, so that concluding here to the superiority of theophylline is quite misleading (the only advantage that could be put forward here is its slow-release formulation). The case report part should therefore be rewritten to make room for all this additional (and crucial) information.

Response: We assume that this reviewer is referring to the publication [Meneret A, Mohammad SS, Cif L, Doummar D, DeGusmao C, Anheim M, et al. Efficacy of Caffeine in ADCY5-Related Dyskinesia: A Retrospective Study. Mov Disord. 2022;37(6):1294-8. Epub 20220405. doi: 10.1002/mds.29006. PubMed PMID: 35384065] that we are also quoting in our manuscript (literature reference [14]). The reviewer is certainly aware that the caffeine dose of 800 mg/day was applied to a 17-year old male patient. Caffeine doses are however not given in mg/kg body weight, which is a deficiency of this publication. It seems obvious that the dose administered to a 17-year-old male patient has to be higher than that applied to a preschool-aged female patient. Therefore, a caffeine dose of 150 mg/day was absolutely sufficient for the patient described herein. Higher doses of caffeine have in fact been tried (up to 3 x 200 mg/day), but no improvement of dyskinesia symptoms was observed any more. Specifically, the patient was never able to walk under caffeine treatment, while under theophylline treatment the patient started walking independently.

The reviewer should also note that in that specific publication (Meneret et al, literature reference [14]) other examples are shown where caffeine was administered to a 15-year-old female patient at 100 mg/day with much improved symptoms and to a 13-year-old female patient at 225 mg/day with very much improved symptoms. This clearly demonstrates that the caffeine dose administered in the patient described herein is not too low.

Question: The authors state in the conclusion that istradefylline exhibits severe side effects. What are their sources? Most reports seem to conclude to limited side effects and acceptable tolerance with this drug. Most reported side effects appear to be dyskinesia, confusion and hallucinations, which are common in a population of Parkinson’s disease patients but cannot be generalized to other patients. Therefore, this needs to be rephrased.

Response: Istradefylline is currently being evaluated at the EMA for approval in the EU to treat Parkinson’s disease. It is known that istradefylline exhibits side effect that have so far hindered approval of this drug in the EU. It is not proven that the side effects of the drug can fully be attributed to common effects that are also observed in Parkinson’s patients. Nevertheless, the respective sentence was rephrased: ”As istradefylline exhibits side-effects even at low-dose application it is currently not an approved drug in Germany.”

---

## [Decision Letter · Decision Letter 1]

23 Jan 2023

PONE-D-22-23902R1Effects of theophylline on ADCY5 activation - From cellular studies to improved therapeutic options for ADCY5-related dyskinesia patientsPLOS ONE

Dear Dr. Sinz,

Thank you for submitting your manuscript to PLOS ONE. After careful consideration, we feel that it has merit but does not fully meet PLOS ONE’s publication criteria as it currently stands. Therefore, we invite you to submit a revised version of the manuscript that addresses the points raised during the review process.

ACADEMIC EDITORPlease indicate your revision and address the comments mainly of Reviewer 2.Best regards

We look forward to receiving your revised manuscript.

Kind regards,

Manuela Cabiati, Ph.D.

Academic Editor

PLOS ONE

Additional Editor Comments:

Dear Prof. Sinz

I apologize for the delay in the review, I hope you be able to answer the question and I am waiting for your revision!

Best regards Manuela cabiati

Reviewers' comments:

Reviewer's Responses to Questions

**Comments to the Author**

1. If the authors have adequately addressed your comments raised in a previous round of review and you feel that this manuscript is now acceptable for publication, you may indicate that here to bypass the “Comments to the Author” section, enter your conflict of interest statement in the “Confidential to Editor” section, and submit your "Accept" recommendation.

Reviewer #1: (No Response)

Reviewer #2: (No Response)

2. Is the manuscript technically sound, and do the data support the conclusions?

Reviewer #1: Partly

Reviewer #2: Yes

3. Has the statistical analysis been performed appropriately and rigorously? 

Reviewer #1: No

Reviewer #2: Yes

4. Have the authors made all data underlying the findings in their manuscript fully available?

Reviewer #1: Yes

Reviewer #2: Yes

5. Is the manuscript presented in an intelligible fashion and written in standard English?

Reviewer #1: Yes

Reviewer #2: Yes

6. Review Comments to the Author

Reviewer #1: The authors have properly addressed some of the issues I raised, but not all of them. I still have concerns:

1) The addition of a sentence concerning the in vitro model is appreciated, but the authors make a statement that is not entirely supported by their results: the clinical results do not “confirm” the cellular results, they add another strong argument in favor of theophylline efficacy in ADCY5-related dyskinesia. It should be at least nuanced as a personal opinion: “we consider that…”

2) The authors have not addressed the second issue properly: they compare the effect of the same concentrations of different molecules that are not equivalent, and show at best tendencies, but no statistically significant differences between molecules. The figures indeed show very high variability of the results, and if we look at the effect of the highest doses used for all 3 molecules, they look about the same. Therefore, based on these results, it is false to draw conclusions on differential effects of these 3 molecules.

3) In addition to the supporting information that has been added, it would be helpful to clearly state in text what the potential side effects of theophylline are (and to cite a paper on the subject). I understand that the authors are themselves well aware of this, but it may not be the case of all readers. It would be very unfortunate if an accident happened due to insufficient monitoring by a prescribing physician who would be less aware of theophylline toxicity. As caffeine is safer, it should be recommended to try it first.

4) There is no weight-dose relationship with caffeine, so that a wide range of doses must be tried in patients before drawing conclusions. The authors have indeed tried higher doses of caffeine in their patient, without better results, which theophylline provided. That has to be said in text, as that information does provide a rationale to try theophylline in patients showing insufficient response to caffeine treatment.

5) As istradefylline has only been tried in Parkinson’s disease patients, its reported side effects may not be generalizable to other populations. The authors should therefore specify that the side effects they are mentioning were found in Parkinson’s disease patients.

Reviewer #2: This is an study reporting on cellular effects of caffeine, theophylline and istradefylline on cAMP production, and the response of a single patient with ADCY5 dyskinesia. The cellular studies are carefully done. The major conclusion is that there was a significant improvement on theophylline exceeding benefit from caffeine.

Given the authors’ emphasis on the clinical improvement of this patient, more care and details about the clinical report and treatment should be provided which are not included in the manuscript nor the supplementary file.

Major comments:

1) State the age of the patient, and report the ADCY5 variant detected in the patient. Was this heterozygous? inherited or de novo?

2) The baseline clinical status of the child prior to theophylline should be included in order to understand the difference with treatment. How far could the child walk with what type of assistance at baseline. What was the scale (0-10 used to assess the improvement?

3) Given the very slow caffeine taper over 5 months, this reviewer surmises at a theophylline dose of 12 mg/kg/day (when improvement is reported), the child was still on both medications. If so, please give the dose of both medications at that time.

4) Please differentiate clearly between parent reports of improvement and assessment made by trained physicians. Eg. Improvement of dyskinesia during sleep is reported by parents only.

5) Were the assessments done by same medical staff repeatedly or different staff over time? Were there attempts at inter-oberver reliability?

6) Was this assessment of the patient in clinic, or by evaluating videos? Were the staff blinded to the treatment (ie did not know the patient, or the dose).

Minor comments

7) In the supplemental file, it would be useful to have a graph tracking the doses of caffeine and theophylline over time, and with time points depicting the theophylline levels, and the improvements.

8) How long has the patient been followed at this point? Are there any long term side effects or waning of the therapeutic effect?

7. PLOS authors have the option to publish the peer review history of their article (what does this mean?). If published, this will include your full peer review and any attached files.

Reviewer #1: No

Reviewer #2: No

---

## [Author Response · Author response to Decision Letter 1]

25 Jan 2023

Responses to Reviewers‘ Comments

Editor:

Question: Please indicate your revision and address the comments mainly of Reviewer 2.

Response: We appreciate the Editor’s suggestion to mainly address the questions raised by reviewer 2. Please note that we have already very carefully and thorougly addressed all comments raised by reviewer 1 in the previous revision of this manusript.

Reviewer 1:

Question: The addition of a sentence concerning the in vitro model is appreciated, but the authors make a statement that is not entirely supported by their results: the clinical results do not “confirm” the cellular results, they add another strong argument in favor of theophylline efficacy in ADCY5-related dyskinesia. It should be at least nuanced as a personal opinion: “we consider that…” 

Response: We replaced the wording “confirms” by “underlines” in the Conclusions section on page 13.

Question: The authors have not addressed the second issue properly: they compare the effect of the same concentrations of different molecules that are not equivalent, and show at best tendencies, but no statistically significant differences between molecules. The figures indeed show very high variability of the results, and if we look at the effect of the highest doses used for all 3 molecules, they look about the same. Therefore, based on these results, it is false to draw conclusions on differential effects of these 3 molecules. 

Response: We believe that our conclusions are fully justified and we have carefully addressed this issue in the previous revision of this manuscript. We agree with the reviewer that according to Figure 3B, the effects of cAMP reduction for caffeine and theophylline are similar to each other after 10 minutes of drug treatment. However, considering the effects of caffeine and theophylline at 60 minutes after drug treatment, the effect of theophylline in respect to cAMP level reduction is more pronounced at a concentration of 100 µM. We have already rephrased the respective sentences in the text accordingly: “The most prominent reduction of cAMP levels was observed for istradefylline, followed by theophylline and caffeine showing a less pronounced cAMP reduction.” “Strikingly, the reduction in cAMP levels was slightly more pronounced for theophylline compared to caffeine.” We believe that this is a correct description of the results obtained.

Question: In addition to the supporting information that has been added, it would be helpful to clearly state in text what the potential side effects of theophylline are (and to cite a paper on the subject). I understand that the authors are themselves well aware of this, but it may not be the case of all readers. It would be very unfortunate if an accident happened due to insufficient monitoring by a prescribing physician who would be less aware of theophylline toxicity. As caffeine is safer, it should be recommended to try it first. 

Response: Again, this point has already been thoroughly addressed in the previous revision of this manuscript. We are fully aware of the narrow therapeutic window of theophylline, which however can be efficiently met by a careful up-dosing of the drug. For this, I have assembled a guideline for medical doctors (and patients) regarding the dosage of theophylline as slow-release formulation for treating patients with ADCY5-related dyskinesia. We have included this guideline in the Supporting Information of this manuscript. Every physician can refer to this guideline. I would like to refer in this context to Paracelsus’ prominent sentence “Dosis facit venenum” every physician learns in their first year of studies. Also caffeine will exhibit severe side effect caused by overdosing.

To clarify this important point, two sentences have already been added on pages 13, 14, and 15 to the previously revised version of this manuscript: “Theophylline exhibits a narrow therapeutic window, which however can be efficiently met by a careful up-dosing of the drug and by regularly monitoring relevant blood parameters (see Supporting Information).” “Several patients are now being treated worldwide with theophylline. In none of the cases, side effects are observed for theophylline when the dosage is slowly increased as described in our dosage scheme and blood parameters are monitored regularly as described (see Supporting Information).” “Theophylline, on the other hand, is available as a slow-release formulation, showing only minor side-effects, even in pediatric applications despite its narrow therapeutic window.”

Question: There is no weight-dose relationship with caffeine, so that a wide range of doses must be tried in patients before drawing conclusions. The authors have indeed tried higher doses of caffeine in their patient, without better results, which theophylline provided. That has to be said in text, as that information does provide a rationale to try theophylline in patients showing insufficient response to caffeine treatment. 

Response: The reviewer should note that the patient described in our study was treated with caffeine in the first place, but the beneficial effects of theophylline have never been observed for caffeine – even when using high doses of caffeine. The same effects, mainly regarding the ability to walk, have in the meantime been observed for theophylline in other patients, too. So it is probably an effect related to theophylline itself that it can specifically improve some of the dyskinesia symptoms better than caffeine.

Question: As istradefylline has only been tried in Parkinson’s disease patients, its reported side effects may not be generalizable to other populations. The authors should therefore specify that the side effects they are mentioning were found in Parkinson’s disease patients.

Response: We have also addressed this point in the previous revision of our manuscript. Istradefylline is currently being evaluated at the EMA for approval in the EU to treat Parkinson’s disease. It is known that istradefylline exhibits side effect that have so far hindered approval of this drug in the EU. It is not proven that the side effects of the drug can fully be attributed to common effects that are also observed in Parkinson’s patients. Nevertheless, the respective sentence was rephrased: ”As istradefylline exhibits side-effects even at low-dose application it is currently not an approved drug in Germany.”

Reviewer 2:

Question: This is an study reporting on cellular effects of caffeine, theophylline and istradefylline on cAMP production, and the response of a single patient with ADCY5 dyskinesia. The cellular studies are carefully done. The major conclusion is that there was a significant improvement on theophylline exceeding benefit from caffeine. 

Response: We appreciate the favorable evaluation of our manuscript by the reviewer.

Question: Given the authors’ emphasis on the clinical improvement of this patient, more care and details about the clinical report and treatment should be provided which are not included in the manuscript nor the supplementary file. 

Response: We have now included information from the clinical report (made by physicians during rehab therapy). The reviewer should note that the report contains sensitive information that touches the patient’s personal rights. Therefore we selected several significant sentences to be included in this manuscript.

Question: State the age of the patient, and report the ADCY5 variant detected in the patient. Was this heterozygous? inherited or de novo? 

Response: On page 11, we have now added additional information on the patient: “A 6-year old female patient with ADCY5-related dyskinesia that had been successfully treated with 150 mg of caffeine (3x 50 mg daily) was treated with 400 mg of slow-release theophylline (2x 200 mg daily).” 

“The patient was diagnosed in 2018 with ADCY5 variant 1252 C>T, resulting in Arg418Trp, heterozygous and de novo mutation, no mosaicism.”

Question: The baseline clinical status of the child prior to theophylline should be included in order to understand the difference with treatment. How far could the child walk with what type of assistance at baseline. What was the scale (0-10) used to assess the improvement? 

Response: We added some sentences from the confidential patient report (rehab report, written by physicians, in German) regarding this issue on page 12: “According to the report of physicians during therapy, the patient was able to make less than 10 steps (with assistance only) and was in a wheelchair during the day. Targeted, slow gripping was possible. Speech was impaired as the patient could speak four to five words and was hard to understand.”

The report was written by physicians (in German) during the rehab therapy of the patient before and during theophylline treatment. As the full report touches the personal rights of the patient it cannot be made available to the public. Please note that we extracted and translated those sentences that are indicative of the patient’s condition during theophylline therapy.

The reviewer should note that on page 12, we have already assessed the improvement of symptoms on a scale (0-10): “On a scale from 0 (no improvement) to 10 (no symptoms), theophylline had a more pronounced influence in the afternoon (6-7), while its influence was ranked between 3-4 in the morning. The rating was done by the medical staff after consultation with the patient’s parents. The duration and frequency of episodes cannot be reliably quantified as different causes (light bacterial or viral infections, stress, joy, etc.) influence motoric conditions during theophylline treatment.”

Question: Given the very slow caffeine taper over 5 months, this reviewer surmises at a theophylline dose of 12 mg/kg/day (when improvement is reported), the child was still on both medications. If so, please give the dose of both medications at that time. 

Response: The exact dosage scheme is summarized in the Supporting Information (dosage of theophylline, slow-release formulation):

“The following dosage scheme has been applied by us:

- A preschool-aged patient with ADCY5-related dyskinesia that had been successfully treated with 150 mg of caffeine (3x 50 mg daily) was treated with 400 mg of slow-release theophylline (2x 200 mg daily).

- Initially, theophylline and caffeine were given in combination, but during the course of the therapy, the daily caffeine dose was reduced over a period of five months until theophylline was administered alone. 

- The theophylline dose was gradually increased from ~6 mg/kg/day to ~23 mg/kg/day over a time course of 8 weeks. Theophylline blood levels ranged between 7.5 mg/l to 21.1 mg/l.

- At a theophylline dose of ~12 mg/kg/day (divided into two single doses), the following effects were observed: The patient became more upright, showed increased muscle tone, and was able to stand and walk independently. The quality of sleep improved as dyskinetic movements subsided completely when falling asleep. 

- By increasing the theophylline dose to ~23 mg/kg/day, continuous improvements were observed.”

Question: Please differentiate clearly between parent reports of improvement and assessment made by trained physicians. Eg. Improvement of dyskinesia during sleep is reported by parents only.

Response: The physicians’ report contains also a statement on sleep quality, which has now been added on page 12: “Under theophylline treatment, the physicians’ report states an improvement in the quality of sleep as dyskinetic movements subsided completely when falling asleep.”

Question: Were the assessments done by same medical staff repeatedly or different staff over time? Were there attempts at inter-oberver reliability?

Response: The assessments of the patient were made by the same team of physicians.

Question: Was this assessment of the patient in clinic, or by evaluating videos? Were the staff blinded to the treatment (ie did not know the patient, or the dose). 

Response: The assessments were always made during stays of the patient in the clinic by direct observation and interaction of the patient with physicians. The staff knew about the patient’s treatment.

Question: In the supplemental file, it would be useful to have a graph tracking the doses of caffeine and theophylline over time, and with time points depicting the theophylline levels, and the improvements. 

Response: We believe that our exact dosage scheme given in the Supporting Information is sufficient:

“The following dosage scheme has been applied by us:

- A preschool-aged patient with ADCY5-related dyskinesia that had been successfully treated with 150 mg of caffeine (3x 50 mg daily) was treated with 400 mg of slow-release theophylline (2x 200 mg daily).

- Initially, theophylline and caffeine were given in combination, but during the course of the therapy, the daily caffeine dose was reduced over a period of five months until theophylline was administered alone. 

- The theophylline dose was gradually increased from ~6 mg/kg/day to ~23 mg/kg/day over a time course of 8 weeks. Theophylline blood levels ranged between 7.5 mg/l to 21.1 mg/l.

- At a theophylline dose of ~12 mg/kg/day (divided into two single doses), the following effects were observed: The patient became more upright, showed increased muscle tone, and was able to stand and walk independently. The quality of sleep improved as dyskinetic movements subsided completely when falling asleep. 

- By increasing the theophylline dose to ~23 mg/kg/day, continuous improvements were observed.”

Question: How long has the patient been followed at this point? Are there any long term side effects or waning of the therapeutic effect?

Response: The patient has been under theophylline treatment for 1.5 years, without any side effects being observed. Also other patients that are now being treated with theophylline did not experience any side effects so far. Please note that one paragraph addressing this issue has been added already during the previous revision of this manuscript: “We did not observe any adverse side effects during theophylline dose titration; arterial blood pressure, pulse rate, and theophylline-related laboratory parameters were within the given reference ranges. Even at the maximum dose of ~23 mg/kg/day, sleep quality was greatly improved with no interruption of sleep for ten hours. Several patients are now being treated worldwide with theophylline. In none of the cases, side effects are observed for theophylline when the dosage is slowly increased as described in our dosage scheme and blood parameters are monitored regularly as described (see Supporting Information).”

---

## [Editor Report · Decision Letter 2]

21 Feb 2023

Effects of theophylline on ADCY5 activation - From cellular studies to improved therapeutic options for ADCY5-related dyskinesia patients

PONE-D-22-23902R2

Dear Dr. Sinz,

We’re pleased to inform you that your manuscript has been judged scientifically suitable for publication and will be formally accepted for publication once it meets all outstanding technical requirements.

Kind regards,

Manuela Cabiati, Ph.D.

Academic Editor

PLOS ONE

Additional Editor Comments (optional):

All my doubts were clarified. the paper is accept in the present form.
---

## [Editor Report · Acceptance letter]

24 Feb 2023

PONE-D-22-23902R2 

Effects of theophylline on ADCY5 activation -
From cellular studies to improved therapeutic options for ADCY5-related dyskinesia patients 

Dear Dr. Sinz:

I'm pleased to inform you that your manuscript has been deemed suitable for publication in PLOS ONE. Congratulations! Your manuscript is now with our production department. 

Kind regards, 

on behalf of

Dr. Manuela Cabiati 

Academic Editor

PLOS ONE